# GenomeBits insight into omicron and delta variants of coronavirus pathogen

**Enrique Canessa**  *, **Livio Tenze**

The Abdus Salam International Centre for Theoretical Physics (ICTP), Trieste, Italy

* canessae@ictp.it

**Data Availability Statement:** Sequence data analyzed here are publicly available at GSIAD https://www.gisaid.org under accession numbers for: [Delta variant]: –Spain: EPI ISL 7900475; EPI ISL 7951710. –USA: EPI ISL 8154562; EPI ISL

## Abstract

We apply the new GenomeBits method to uncover underlying genomic features of omicron and delta coronavirus variants. This is a statistical algorithm whose salient feature is to map the nucleotide bases into a finite alternating (±) sum series of distributed terms of binary (0,1) indicators. We show how by this method, distinctive signals can be uncovered out of the intrinsic data organization of amino acid progressions along their base positions. Results reveal a sort of *'ordered'* (or constant) to *'disordered'* (or peaked) transition around the coronavirus S-spike protein region. Together with our previous results for past variants of coronavirus: Alpha, Beta, Gamma, Epsilon and Eta, we conclude that the mapping into GenomeBits strands of omicron and delta variants can help to characterize mutant pathogens.

## Introduction

Since the coronavirus outbreak in Wuhan, China, in December 2019, the SARS-CoV-2 pandemic has became a major risk in global public health. The impact of the outbreak on access to healthcare services has left important repercussions. Severe effects on the mental health and well-being of medical staff and people around the world have had also a lot of relevant implications [1]. Many other sectors such as world economic systems have also suffered significantly due to the induced Covid-19 restrictions.

Understanding the coronavirus pathogen is still a global challenge for scientific research. The identification by Similarity studies and fast genomic analysis of the positive-stranded RNA virus –continuously provided through the complete genome sequences confirmed from different laboratories around the world, allowed to shed light into the evolutionary origins of SARS-CoV-2 lineage [2]. These studies suggested that the SARS-CoV-2 genome may be in fact formed via recombination of genomes close to the RaTG13 and GD Pangolin CoV genomes, and be a close relative of bat CoV ZC45 and ZXC21. However, the intermediate source species of SARS-CoV-2 have not been confirmed so far (for a updated preview see Ref [3]). The importance of understanding the origin of this coronavirus, natural or unnatural, will help to prevent possible future pandemics. This debate, however, highlights the need for a global network of real-time surveillance systems, with the capacity to rapidly deploy genomic tools and statistical studies for pathogen identification and characterization of the evolution of potential

8178972; EPI ISL 8179449. [Omicron variant]:
–Spain: EPI ISL 8170071; EPI ISL 8170084. –USA:
EPI ISL 8163263; EPI ISL 8163267; EPI ISL
8163304; EPI ISL 8163380; EPI ISL 8082518; and
[Original sequence]: –Wuhan-Hu-1 China:
MN908947.3 from GenBank. See also: https://
github.com/canessae/GenomeBits/.

**Funding:** The author(s) received no specific
funding for this work.

**Competing interests:** The authors have declared
that no competing interests exist.

lineages and mutations. SARS-CoV-2 is still spreading worldwide with evolving variants, some of which occur in the Spike protein and appear to increase viral infection [4]. The recent study in Ref [4], emphasizes the need to track and analyze viral sequences also in relation to clinical status.

To this last aim we consider in this work, and apply anew, the GenomeBits method [5]. This is a simple statistical algorithm whose salient feature is to map the nucleotide bases A,C, G,T into a finite alternating (±) sum series of distributed terms of binary (0,1) indicators. The method can provide additional information to conventional comparative Similarity studies via alignment methods [6], specially on single nucleotide structures to detect the effects of mutations. That is, within the single nucleotide polymorphisms of a genome sequence. These polymorphisms reveal the complex dynamic gene transfer-recombination events among lineages with potential to cause major human outbreaks [7]. We report here a kind of '*ordered*' (or constant) to '*disordered*' (or peaked) transition around the S-spike protein region.

Previously by GenomeBits [5], we uncovered distinctive signals for the intrinsic gene insights in the coronavirus genome sequences for past variants of concern: Alpha, Beta and Gamma, and the variants of interest: Epsilon and Eta. We enhance here this previous study of GenomeBits into the most recent delta and omicron variants of the coronavirus pathogen. The first known confirmed delta variant was detected in India in late 2020 and the B.1.1.529 infection appeared from a specimen collected in South Africa a year later, early November 2021 [8, 9]. Omicron variants have grown dominant world-wide and seems to be continuously evolving. Evolution selects those mutations that replicate more efficiently.

The delta and omicron variants share some parts of their structures [10, 11]. Both variants have common mutations, *i.e.*, amino acid changes in the building blocks that conform the spike protein (D614G mutation), and different mutations elsewhere (the P323L mutation in the NSP12 polymerase and the C241U nucleotide mutation in the 5' untranslated region). The latter seems to give a greater advantage for their replicating capacity. Omicron seems to cause less severe COVID-19 than delta as seen from data on duration of hospital stays, ICU admittance and deaths which are reported lower than during previous pandemic waves [12].

The ongoing SARS-CoV-2 research is currently focusing on understanding the essential functions of the conforming proteins in the ribonucleic acid RNA coronaviruses [13]. These consists of an unusually large collection of RNA synthesizing and processing enzymes to express and replicate genome sequences that are targets for antiviral drug design. One possibility to unravel genome organization and variants of viruses is through statistical approaches as shown by the present results.

## Comparison methods

Genome sequence comparisons and the discovery of new signaling pathways can be analyzed using the GenomeBits method [5, 14]. We apply GenomeBits to uncover distinctive patterns from delta and omicron genome sequences available in the GISAID archive at www.gisaid.org. As discussed below, this method provides additional information to conventional approaches via Similarity comparisons (for a comparison with Fourier Power Spectrum studies see [5]).

### Similarity plots

Similarity between pairs of full-length genome sequences is the standard method for determining whether there are sequence equivalences in terms of shared ancestry between them by using alignment methods [6, 15]. We show in Fig 1, genetic Similarity plots of different query sequences of SARS-CoV-2 genome. We downloaded genome sequence data in FASTA format collected in December 2021 (whose GISAID accession IDs are also indicated in the figures).

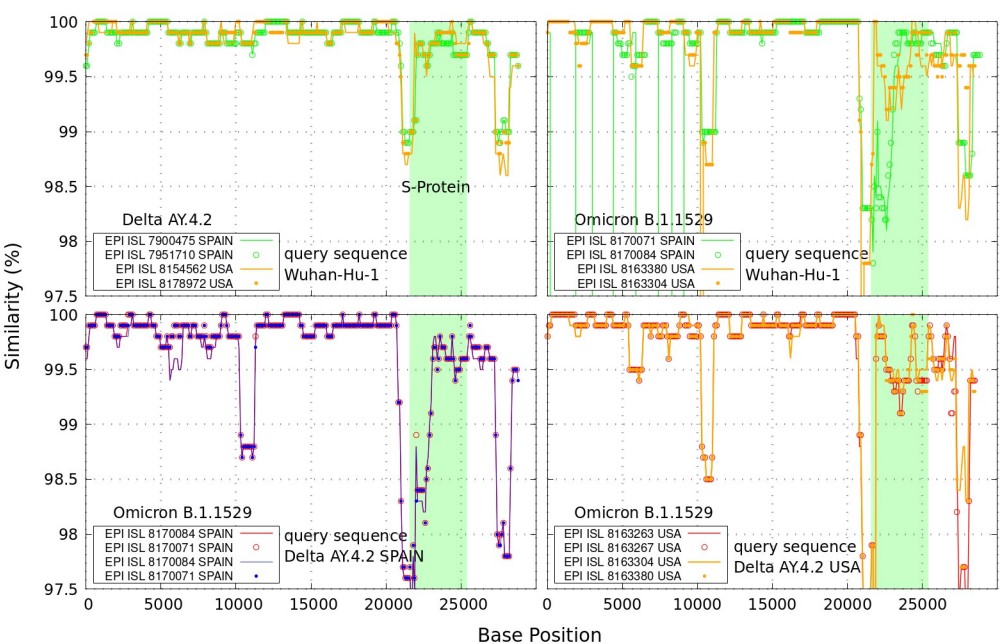

**Fig 1. Similarity plots.** Upper curves: genetic similarity curves between the query sequence SARS-CoV-2 Wuhan-Hu-1 and representative delta and omicron complete genome sequences. In clear blue is the genomic region encoding the spike (S-protein). Lower curves: delta genome sequences used as query against omicron data from Spain and USA. A typical sliding 1000 base pair window in steps of 100 nucleotide bases position was used in these calculations.

The FASTA format is the commonly used text-based format for representing nucleotide sequences in which amino acids are represented by single-letter codes: (A)denine, (C)ytosine, (G)uanine and (T)hymine (or Uracil RNA genome for single strand folded onto itself). They store the instructions to assemble and reproduce every living organism.

By the Similarity plot, we verified more deviations of omicron variants than delta variants from Spain and USA with respect to the first Wuhan-China sequences identified over a year ago (MN908947) [16]. The lower curves in Fig 1 display larger deviations of delta variants from Spain (EPI ISL 7900475 and 7951710) against omicron variants from Spain, as well as the delta variant from USA (EPI ISL 8179449) against omicron variants from USA. In these calculations we used "lalign36" sequence comparison software via the Waterman–Eggert algorithm at http://github.com/wrpearson/fasta36 [5].

Conventional Similarity comparisons via "lalign36" alignment provides limited information on the single nucleotide bases A,C,G,T. To determine the best parameters to achieve optimal alignments is difficult. There are several user-defined parameters to overcome gaps and mismatches usually found between genome sequences. Furthermore, the computational resources required increase considerably depending on the length and number of sequences to be aligned.

In the figure regions with clustering of SARS-CoV-2 sequences (< 1%) from the city of Wuhan in relation to the delta lineages from SPAIN, suggest some genetic similarities outside the S-spike gene region (bp 21563–25384, colored in clear blue). More divergent genetic similarities (∼ 97%) are found between the first Wuhan-China sequences and omicron strains from Spain and USA, and in between the delta sequences from Spain and USA against omicron variants from Spain and USA, respectively. This is a clear consequence of the coronavirus mutations.

## GenomeBits representation

Our new quantitative method for the examination of distinctive patterns of complete genome sequences considers a certain type of alternating series having terms converted to (0,1) binary values for the nucleotide variables $\alpha = A, C, T, G$ as observed along the reported genome sequences, namely

$$E_{\alpha,N}(X) = \sum_{k=1}^{N} (-1)^{k-1} X_{\alpha,k} \, , \qquad (1)$$

where the individual terms $X_k$ are associated with 0 or 1 values according to their position along the genome sequences of length $N$, satisfying the following relation

$$X_{\alpha,k=N} = \left| E_{\alpha,N}(X) - E_{\alpha,N-1}(X) \right| \, . \qquad (2)$$

The arithmetic progression carries positive and negative signs $(-1)^{k-1}$ and a finite non-zero first moment of the independently distributed variables $X_{k,\alpha}$.

The mapping into four binary projections of genome sequences follows previous studies on the three-base periodicity characteristic of protein-coding DNA sequences [17]. However, as a principal difference with other binary representations (see also, *e.g.*, Refs [18, 19]), in our mapping the terms in the sums change sign. If a term $X_{\alpha,k}$ is positive at a given nucleotide base position (bp) $k$, then the successive $X_{\alpha,k+1}$ term is negative and vice versa. This selection is inspired by the well-know model of Ising spins model in Physics in which discrete variables representing magnetic dipole moments of atomic *spins* in a lattice can be in one of two states: +1 or −1. In our case the binary indicator for the sequences carries alternating ± signs where plus and minus signs are chosen sequentially starting with +1 at $k = 1$ as illustrated in the example of Table 1. In the Table a mapping example for converting the brief genome fragment AGATCTGTTCTC (consisting of 12 nucleotides) into the alternating binary array via Eq (1) is given.

There is a user-friendly Graphics User Interface (GUI) to the present signal analysis method of genome sequences. The GUI runs under Linux Ubuntu O.S. and can be downloaded freely from Github [14]. It is efficient and requires little processing time for large genomic data. Our GenomeBits GUI considers samples with A,C,T,G sequences for (up to two) given Countries corresponding to genomic sequence data from (up to six) given variants/species. It discards uncompleted sequences containing codification errors (usually denoted with "NNNNN" and other letters).

In brief, with just one click the GenomeBits GUI allows to

**Table 1. Example of genome sequence-to-GenomeBits mapping (via Eq (1) for $N = 12$).**

| Base position $k$ | 1 | 2 | 3 | 4 | 5 | 6 | 7 | 8 | 9 | 10 | 11 | 12 | GenomeBits sums |
|---|---|---|---|---|---|---|---|---|---|---|---|---|---|
| Sequence (string) | A | G | A | T | C | T | G | T | T | C | T | C | $\sum_{k=1}^{12} (-1)^{k-1} X_{\alpha,k}$ |
| $(-1)^{k-1} X_{\alpha=A,k}$ | +1 | 0 | +1 | 0 | 0 | 0 | 0 | 0 | 0 | 0 | 0 | 0 | 2 |
| $(-1)^{k-1} X_{\alpha=C,k}$ | 0 | 0 | 0 | 0 | +1 | 0 | 0 | 0 | 0 | -1 | 0 | -1 | -1 |
| $(-1)^{k-1} X_{\alpha=G,k}$ | 0 | -1 | 0 | 0 | 0 | 0 | +1 | 0 | 0 | 0 | 0 | 0 | 0 |
| $(-1)^{k-1} X_{\alpha=T,k}$ | 0 | 0 | 0 | -1 | 0 | -1 | 0 | -1 | +1 | 0 | +1 | 0 | -1 |

The variable $\alpha$ represents single-letter nucleotide codes: (**A**)denine, (**C**)ytosine, (**G**)uanine and (**T**)hymine. Within the GenomeBits method, the ± signs are chosen sequentially starting with plus at the nucleotide base position $k = 1$ by default.

- run the alternating sums in Eq (1) for up to six-times-two inputs of FASTA files containing (*i.e.*, concatenating) more than one genome sequence each;

- separate concatenated genome sequences and save it in single FASTA files (for each country);

- get into single files, each of the four nucleotide bases represented by the symbols A,C,T,G;

- get the alternating sums results in single files for each of the four nucleotide bases A,C,T,G associated with (±) binary values;

- plot the alternating sums to compare behavior of pairs A,T and C,G nucleotide bases versus nucleotide bp;

- compare in a plot alternating sums curves versus bp for all four nucleotide bases A,C,T,G;

- plot the alternating sums curves versus bp for each of the four nucleotide bases A,C,T,G;

- plot results for each nucleotide base A,C,T or G for up to six variants/species and up to 4 FASTA files by country;

- comparison of GenomeBits GUI curves for (up to six) given variants/species and (up to two) selected Countries, with those results from our original paper in [5].

We shall show next that analyzing genomic sequencing via the present type of finite alternating sums allows to extract unique features for omicron and delta mutations with little data noise variations. From the viewpoint of statistics, such series are equivalent to a discrete-valued time series for the statistical identification and characterization of (random) data sets [20].

## Discussion

The GenomeBits representation of coronavirus genome variants, by adding binary values with ± signs following Eq (1), can reveal interesting imprints of the genome dynamics at the level of nucleotide ordering. By this method of binary projections we are able to uncover distinctive signals of the intrinsic gene organization embedded in the genome sequences of the single-stranded RNA coronaviruses. This approach allow to judge nucleotide bases mutations between omicron and delta genome sequences by simply alternating their associated (±) binary coding as illustrated in Table 1.

In Fig 2, we show the results obtained for the sequences of each A, C, G and T nucleotide of the coronavirus variants of concern—AY.4.2 (delta) and B.1.1529 (omicron) reported from Spain and the USA for a number of representative samples as indicated. As reference, we display on the left of the plot our results for nucleotides A,C of the strand and on the right the nucleotides T,G ("complementary to those of the opposite strand"—according to the pairing rules A-T and C-G of DNA). The complete genome sequences consists of $N$ nucleotides on the order of 30,000 base pairs in length, two to three times larger than that of most other RNA viruses [13].

It is interesting to note how in the figure there are regions where the curves for the delta variant (in blue) mirror those of the omicron variant (in green). This peculiar behavior becomes clear by averaging both curves as shown by the red lines. The regions of almost null (with low data noise), or rather constant average values, indicates rather perfect mirroring matching, which is driven by the ± signs of the alternating series. This reveals coding regions of correspondence between delta and omicron variants.

The regions of main discrepancies as found in the Similarity identities curves of Fig 1, e.g., around $N = 10000$ are also reflected by the red lines of Fig 2. The main difference between

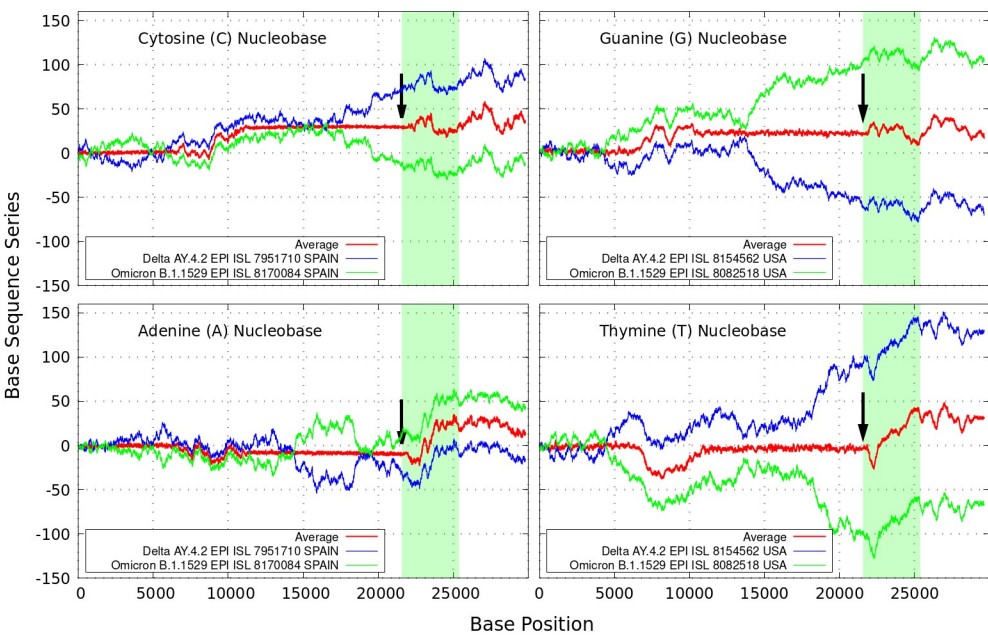

**Fig 2. Sequence sum series.** Delta (in blue) and omicron (in green) variant imprints displayed by the nucleotides A,C, G,T according to Eq (1) along different samples of the genomic strand of coronavirus available from Spain and USA. The arrows indicate a sort of *'ordered'* (constant) to *'disordered'* (peaked) transition before the coding region of the S-spike genes for the SARS-CoV-2 Wuhan-Hu-1 sequence (drawn in clear blue).

both comparative genomic approaches is that changes via Eq (1) can be analyzed and characterized at each single A,C,G,T nucleotide level. This is important because single nucleotide polymorphisms are the most common genomic variations. The figure illustrate a kind of *'ordered'* (constant) to *'disordered'* (peaked) transition phenomena around the NSP-5 polymerase within the open reading frames ORF1a region [10, 11], up to the nucleotide region of the S-Protein (colored clear blue area).

To some degree, there are also other distinctive trends especially around the S-Protein. As seen in Fig 2, the black arrows indicate a phase transition point appearing close to the coding region of the S-spike genes. The peaked curves diverge rapidly and tend to separate denoting bigger dissimilarities for increasing $N$. It is worth noting that the patterns for the base sequence series for Adenine and Cytosine display completely different convergences between the variants considered. The positive and negative terms in the sums of Eq (1) for the discrete $\alpha$ variables, partly cancel out allowing the series "to converge" to some non-zero values for all the nucleotide classes. At the black arrows positions, this feature allows to estimate a separation ratio of $\sim 2$ between nucleotide C and G, and $\sim 4$ times higher between the curves for A and T. We also note that the inclusion of different smoothing sliding window sizes of up to about 500 bp (moving along the target genome sequences and repeating the GenomeBits procedure as described, lead to a data noise reduction in the curves and preserve the average behavior of the sums displayed in Figs 1 and 2.

## Conclusions

We have applied the GenomeBits method Eq (1) to uncover underlying genomic features of omicron and delta coronavirus variants. By this method, we have found a sort of *'ordered'* (or constant) to *'disordered'* (or peaked) transition around the coronavirus S-spike protein region.

This result is significant because it has been obtained by assigning binary strings to symbolic nucleotide characters. Via this simple assignment, one can compare various gene sequences as done so far. Consequently, the present nucleotide mapping representation may be useful to archive the dynamical (history) properties of the original sequence.

Numerical representations of genome sequences have gained great attention in bioinformatics studies. One advantage for this approach is that large sequence data can be handled statistically to find various characterizations. Additional properties of the genome sequences for mutant pathogens, as derived in this work for an *'ordered'*-to-*'disordered'* transition, may also allow to locate and distinguish polymers of amino acids (proteins states and positions) in a sequence and determine if the altered genes may behave similar to those already targeted.

GenomeBits may shed light on the bioinformatics surveillance behind future infectious diseases. By a comparison of numerical results, it may be also of some relevance to assist in further developments of synthetic mRNA-based vaccine designs [13, 21]. Such comparative genomic statistical representations can offer insights on the inherent data organization during the natural evolution of pandemic. Letter sequence-to-numerical signal mappings are likely to continue in future genomic encodings of new sequences.

## Supporting information

**S1 Text.**
(TEX)

## Acknowledgments

We thank all authors and Labs who have kindly deposited and shared genome data on GISAID. We are also indebted to the academic editor and the two reviewers for providing insightful comments on this manuscript.

## Author Contributions

**Conceptualization:** Enrique Canessa.

**Formal analysis:** Enrique Canessa.

**Investigation:** Enrique Canessa.

**Methodology:** Enrique Canessa.

**Project administration:** Enrique Canessa.

**Software:** Enrique Canessa, Livio Tenze.

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
