## [Decision Letter · Decision Letter 0]

27 Apr 2022

PONE-D-22-01925GenomeBits insight into omicron and delta variants of coronavirus pathogenPLOS ONE

Dear Dr. Canessa,

Thank you for submitting your manuscript to PLOS ONE. After careful consideration, we feel that it has merit but does not fully meet PLOS ONE’s publication criteria as it currently stands. Therefore, we invite you to submit a revised version of the manuscript that addresses the points raised during the review process.

We look forward to receiving your revised manuscript.

Kind regards,

Vladimir Makarenkov

Academic Editor

PLOS ONE

Journal Requirements:

Additional Editor Comments:

It would be important to add a discussion (preferably to the Introduction section) about the origins and impact of the SARS-Cov-2 pandemic. Here you could cite the following works:

Boni, Maciej F., et al. "Evolutionary origins of the SARS-CoV-2 sarbecovirus lineage responsible for the COVID-19 pandemic." Nature microbiology 5.11 (2020): 1408-1417.

Makarenkov, V., Mazoure, B., Rabusseau, G. et al. Horizontal gene transfer and recombination analysis of SARS-CoV-2 genes helps discover its close relatives and shed light on its origin. BMC Ecol Evo 21, 5 (2021).

Domingo JL. What we know and what we need to know about the origin of SARS-CoV-2. Environ Res. 2021;200:111785.

Reviewers' comments:

Reviewer's Responses to Questions

**Comments to the Author**

1. Is the manuscript technically sound, and do the data support the conclusions?

Reviewer #1: Partly

Reviewer #2: Partly

2. Has the statistical analysis been performed appropriately and rigorously? 

Reviewer #1: N/A

Reviewer #2: Yes

3. Have the authors made all data underlying the findings in their manuscript fully available?

Reviewer #1: Yes

Reviewer #2: Yes

4. Is the manuscript presented in an intelligible fashion and written in standard English?

Reviewer #1: No

Reviewer #2: No

5. Review Comments to the Author

Reviewer #1: 1. The write-up of the manuscript seems to be premature and very difficult to understand. There are several sweeping statements without any references. For instance, the statement “The first known confirmed delta variant was detected in India in late 2020 and the B.1.1.529 infection appeared from a specimen collected in South Africa a year later, early November 2021” should have a citation. Several sentences are incomplete or do not have a meaning. For example “Although omicron seems to cause less severe COVID-19 than delta.”

2. Authors should clearly explain the computation of binary scores for a given genomic sequence. Illustration or outline graphic may be required.

3. It is trivial that a mononucleotide composition can reveal similar information. How Genomebit information is different from mononucleotide computation with a sliding window?.

4. There is no discussion on their results. The authors should compare their analysis with their previous work and also with similar research by others.

Reviewer #2: The manuscript on the topic GenomeBits insight into omicron and delta variants of coronavirus pathogen is an interesting research article. The manuscript is with the interest to the reader and fully in the scope of journal.

I will suggest the manuscript to be accepted for publication after revision.

1. Abstract section looks incomplete. I will suggest the author to focus on following important points on writing the abstract. An abstract summarizes, usually in one paragraph of 300 words or less, the major aspects of the entire paper in a prescribed sequence that includes: 1) the overall purpose of the study and the research problem(s) you investigated; 2) the basic design of the study; 3) major findings or trends found as a result of your analysis; and, 4) a brief summary of your interpretations and conclusions.

2. English of the script is very poorly written. Please write your text in good English (American or British usage is accepted, but not a mixture of these). English language manuscript may require editing to eliminate possible grammatical or spelling errors and to conform to correct scientific English.

3. There are several sentences in the script which are really hard to understand. I will suggest the authors should carefully read the script and amend the English language correction throughout the script.

4. Introduction section need to be more elaborated.

5. Provide a general interpretation of the results in the context of other evidence, and implications for future research.

6. PLOS authors have the option to publish the peer review history of their article (what does this mean?). If published, this will include your full peer review and any attached files.

Reviewer #1: **Yes: **Dr Kishore Sesham

Reviewer #2: **Yes: **SADANAND PANDEY

---

## [Author Response · Author response to Decision Letter 0]

13 May 2022

See also attached file "Response_to_Reviewers.pdf" for our reply to each specific reviewer and editor comments.

Response to Reviewers of PONE-D-22-01925

“GenomeBits insight into omicron and delta variants of coronavirus pathogen”,

by E. Canessa, L. Tenze

We are indebted to the academic editor and the two reviewers for providing

insightful comments on the manuscript. Our responds to each point raised are

as follows:

Reply to Additional Editor Comments:

The Introduction section was revised and completely rewritten. In particular, we

added a discussion in the Introduction section about the origins of the SARSCov-

2 pandemic as sugested by the Editor, and cited the three works

suggested: (i) Boni, Maciej F., et al. (ii) Makarenkov, V., Mazoure, B.,

Rabusseau, G. et al., and (iii) Domingo JL. We also briefly mentioned the

impact of the SARS-Cov-2 pandemic and added 11 new references to support

our findings throughout the text. The abstract and conclusions have also been

improved based on the data presented. All sequence data analysed here are

publicly available at GSIAD and all custom code used in the manuscript is fully

available at https://github.com/canessae/GenomeBits/ We have made an effort

to present our manuscript in an intelligible fashion with an standard English. At

revision, we corrected few typographical and grammatical errors.

Reply to Reviewer #1:

1. We have made an effort to rewrite our manuscript and present it in an

intelligible fashion with an standard English. Our statements now carry out

associated new references and few sentences have been completed with a

clear meaning.

2. We have added a new Table to aIllustrate the computation of binary scores

for a given genomic sequence. This Table helps us to explain that in our case,

the mapping into four binary projections of genome sequences carries

alternating plus and minus signs, where + and - signs are chosen sequentially

starting with +1 at k = 1 as default. The Table shows a particular mapping

1

example for converting the brief fragment AGATCTGTTCTC of 12 nucleotides

into the alternating binary array. As a principal difference with previous binary

representations (see, e.g., new Refs [19, 20]), our mapping (whose terms

change sign –i.e., if a term X_k is positive then X_k+1 is negative and vice

versa) displays their occurrence as +1 or -1 as well as their non existence as 0

at a given base pair k.

3. The inclusion of different smoothing sliding window sizes of up to about 500

bp (moving along the target genome sequences and repeating the GenomeBits

procedure as described, lead to a data noise reduction in the curves and

preserve the average behavior of the sums displayed in Figures 1 and 2.

4. Our results have been further elaborated. We emphasize that the regions of

main discrepancies as found in the Similarity identities curves of Fig 1, e.g.,

around N=10000 are also reflected by the red lines of Fig 2 via GenomeBits.

The main difference between both comparative genomics approaches is that

changes via Eq (1) can be analyzed and characterized at each single A,C,G,T

nucleotide level separately. Beside such comparision of Similarity studies and

our previous analysis on Fast Fourier Transforms for coronavirus genome of

other variants as reported in Ref[6], our present findings are new and have not

similars to those reported by others. We report a kind of 'ordered' (constant) to

'disordered' (peaked) phase transition phenomena around the NSP5

polymerase within the open reading frames ORF1a region, up to the nucleotide

region of the S-Protein. As seen in Fig 2, the black arrows indicate an

analogous phase transition point appearing close to the coding region of the Sspike

genes.

Reply to Reviewer #2:

We particularly thanks this referee for finding our research article interesting

and fully in the scope of PLOS One journal.

1. As suggested, the new Abstract now reads: “We apply the new GenomeBits

method to uncover underlying genomic features of omicron and delta

2

coronavirus variants. This is a statistical algorithm whose salient feature is to

map the nucleotide bases into a finite alternating (+-) sum series of distributed

terms of binary (0,1) indicators. We show how by this method, distinctive

signals can be uncovered out of the intrinsic data organization of amino acid

progressions along their base positions. Results show a sort of 'ordered' (or

constant) to 'disordered' (or peaked) transition around the coronavirus S-spike

protein region. Together with our previous results for past variants of

coronavirus: Alpha, Beta, Gamma, Epsilon and Eta, we conclude that the

mapping into GenomeBits strands of omicron and delta variants can help to

characterize mutant pathogens.”

2. We have made an effort to present our manuscript in an intelligible fashion

with an standard English. At revision, we corrected few typographical and

grammatical errors.

3. (idem as 2)

4. As also replied to the Academic Editor, the Introduction section was revised

and completely rewritten. In particular, we added a discussion in the

Introduction section about the origins of the SARS-Cov-2 pandemic and cited

several new references. The abstract and conclusions have also been improved

based on the data presented

5. In the Conclusion section we now provide a general reinterpretation of the

results in the context of the present study and other evidence published, and

also briefly discussed implications for future research.

-oOo-

---

## [Decision Letter · Decision Letter 1]

23 Jun 2022

GenomeBits insight into omicron and delta variants of coronavirus pathogen

PONE-D-22-01925R1

Dear Dr. Canessa,

We’re pleased to inform you that your manuscript has been judged scientifically suitable for publication and will be formally accepted for publication once it meets all outstanding technical requirements.

Kind regards,

Vladimir Makarenkov

Academic Editor

PLOS ONE

Additional Editor Comments (optional):

Reviewers' comments:

Reviewer's Responses to Questions

**Comments to the Author**

1. If the authors have adequately addressed your comments raised in a previous round of review and you feel that this manuscript is now acceptable for publication, you may indicate that here to bypass the “Comments to the Author” section, enter your conflict of interest statement in the “Confidential to Editor” section, and submit your "Accept" recommendation.

Reviewer #2: All comments have been addressed

2. Is the manuscript technically sound, and do the data support the conclusions?

Reviewer #2: Yes

3. Has the statistical analysis been performed appropriately and rigorously? 

Reviewer #2: N/A

4. Have the authors made all data underlying the findings in their manuscript fully available?

Reviewer #2: Yes

5. Is the manuscript presented in an intelligible fashion and written in standard English?

Reviewer #2: Yes

6. Review Comments to the Author

Reviewer #2: (No Response)

7. PLOS authors have the option to publish the peer review history of their article (what does this mean?). If published, this will include your full peer review and any attached files.

Reviewer #2: **Yes: **SADANAND PANDEY

---

## [Editor Report · Acceptance letter]

1 Jul 2022

PONE-D-22-01925R1 

GenomeBits insight into omicron and delta variants of coronavirus pathogen 

Dear Dr. Canessa:

I'm pleased to inform you that your manuscript has been deemed suitable for publication in PLOS ONE. Congratulations! Your manuscript is now with our production department. 

Kind regards, 

on behalf of

Dr. Vladimir Makarenkov 

Academic Editor

PLOS ONE